# Body-Related Visual Biasing Affects Accuracy of Reaching

**DOI:** 10.3390/brainsci14121270

**Published:** 2024-12-17

**Authors:** Claude Beazley, Stefano Giannoni, Silvio Ionta

**Affiliations:** 1SensoriMotorLab, Department of Ophthalmology-University of Lausanne, Jules Gonin Eye Hospital-Fondation Asile des Aveugles, 1004 Lausanne, Switzerland; claude.beazley@fa2.ch (C.B.); sgiannonil@gmail.com (S.G.); 2Centre de compétences pour le déficit visuel (CPHV), 1004 Lausanne, Switzerland

**Keywords:** eye–hand coordination, visuomotor integration, movement, reaching, distortion, adaptation

## Abstract

**Background:** Many daily activities depend on visual inputs to improve motor accuracy and minimize errors. Reaching tasks present an ecological framework for examining these visuomotor interactions, but our comprehension of how different amounts of visual input affect motor outputs is still limited. The present study fills this gap, exploring how hand-related visual bias affects motor performance in a reaching task (to draw a line between two dots). **Methods:** Our setup allowed us to show and hide the visual feedback related to the hand position (cursor of a computer mouse), which was further disentangled from the visual input related to the task (tip of the line). **Results:** Data from 53 neurotypical participants indicated that, when the hand-related visual cue was visible and disentangled from the task-related visual cue, accommodating movements in response to spatial distortions were less accurate than when the visual cue was absent. **Conclusions:** We interpret these findings with reference to the concepts of motor affordance of visual cues, shifts between internally- and externally-oriented cognitive strategies to perform movements, and body-related reference frames.

## 1. Introduction

Although apparently simple, moving the cursor of the computer mouse to click an icon is a rather complex task. It requires accurately using the visual inputs about both the target and the cursor to move the hand/mouse and to keep continuously monitoring the output of the movement until the cursor hits the target. This act of reaching involves extensive use of visual inputs to (i) assess where the mouse cursor is in relation to the target, (ii) compute in which direction the user should move the cursor, and (iii) estimate whether the moving cursor would likely hit or miss the target. This visual information must be constantly updated and fed back to the motor system so that online correction of movement can be eventually applied to reach the goal. Visuomotor psychophysics focuses on these complex visuomotor interactions during actions like throwing and catching things [1], writing [2], using objects [3,4], and moving a cursor on a screen [5]. Previous work showed that continuous visual feedback about the outcome of the movement is necessary to perform accurate movements [6,7,8,9,10,11]. For instance, visually impaired children can present impaired visuomotor skills in the absence of basic motor deficits [12,13,14]. A steadily growing body of behavioral evidence supports the involvement of the visual system in visuomotor performance [15,16]. Altering visual elements like scaling factors and density of optical flow can impact task performance and motor control strategies [17]. Continuous visual feedback enhances performance when closed-loop control is feasible, but it can hinder performance once visual input is removed [18]. During reaching, visual estimates are integrated into motor planning in order to predict the precision of potential movement and therefore balance viewing time and movement duration to enhance visuomotor accuracy [19]. These bindings between visual inputs and motor outputs are further supported by clinical observations that, for instance, damage to the dorsal stream leads to visuomotor deficits like optic ataxia [20]. Optic ataxia is characterized by the inability to perform precise, visually guided movements despite having no general motor impairments [21,22]. It is a typical visuomotor integration disorder [23], linked to visuomotor neurons located both in visual and motor brain regions [24]. Furthermore, the development of visuomotor skills has been associated with basic visual functions, such as global motion perception [25,26] and depth perception [27,28], and from the first months of life, children continuously reach and grasp any objects in their reaching space [29].

At the neural level, animal research showed that plasticity in the visual cortex is essential for visuomotor performance [30], the activity of the visual cortex can drive the timing of subsequent movements [31], and reaction times in oculomotor tasks can be correlated with the activation of the visual cortex [32,33]. In humans, similarly straightforward demonstrations of the involvement of the visual cortex in visuomotor performance are rarer. On the one hand, it has been reported that the visual cortex can interact with the motor network to enhance accurate motor performance [34], variations in visual input can affect the excitability of the motor cortex [35], and that the activity of the visual cortex correlates with the response times in psychophysical tasks [36] and visuomotor skills [37] tasks. On the other hand, other studies reported that visuomotor performance is independent of alpha-band frequency [38], which can be considered an index of the activity in the visual cortex [39]. Thus, definitive conclusions cannot be drawn.

Visuomotor interactions have been studied mostly through the analysis of visually guided movements such as reaching, in that participants are asked to place their hand/finger in a given position (usually holding a computer mouse) and to reach targets in different positions (indicated by dots on a computer screen). This approach showed that simultaneous visual and proprioceptive information is crucial for visuomotor adaptation [40]. To assess the influence of visual input on motor output, many studies manipulated the properties of the visual input. For instance, Izawa and Shadmehr [41] intervened on the clarity (blur) of the targets while participants were executing reaching movements and found that the precision of reaching movements was directly proportional to the clarity of the target. Handlovsky et al. [42] varied the perceived size of the targets and showed that reaching movements were faster when the target appeared larger. Veerman et al. [43] reported that adjustments in the trajectory and velocity of reaching movements occurred in response to altered properties of the targets. While these studies support the tight link between vision and movements, their experimental approach was not designed to disentangle whether achieving the task relied more on the visual or proprioceptive inputs related to the hand movement, because the visual feedback (e.g., cursor) was always visible and corresponded to the position of the mouse/hand (proprioception). Another way to test the impact of visual input on motor output is to introduce visual distortions to disrupt the relationship between the expected and the real visual outcome of a movement. With this approach, Saijo et al. [44] showed that when the entire visual field is shifted toward one side, the hand movement deviates towards the same side; Whitney et al. [45] found that shifting the background of a visual scene influenced the trajectory of the hand movements, and Brenner and Smeets [46] observed that movements are adjusted to accommodate shifts in the visual environment even when the target is stationary. Since in these setups the visual distortions affect both the visual output of the movement (e.g., drawing a line) and the visual representation of the proprioceptive feedback from the hand (e.g., cursor), these experimental approaches do not allow us to disentangle whether the obtained effects are driven by visuo-visual or visuo-proprioceptive conflicts.

Thus, neither modifications of the visual appearance nor visuospatial distortions of visuospatial feedback can disentangle the role of movement-related visual and proprioceptive feedback during reaching or other visuomotor interactions. To fill this gap, the present study investigated the relationship between vision and proprioception in the context of visuomotor interactions. To this aim, we assessed whether a visuomotor task (line drawing) is influenced by the visual feedback (cursor) of proprioceptive input (hand), measuring how the performance in a reaching task changes in presence and absence of the cursor. We hypothesized that, when hand-related visual cues (e.g., cursor) are disentangled from task-related visual cues (e.g., line tip), participants would exhibit different accuracy in accommodating visual distortions in presence versus absence of hand-related visual cues.

## 2. Materials and Methods

### 2.1. Participants

A total of 56 healthy young adults signed up for the experiment and were screened according to the following inclusion criteria: age between 18 and 30 years, right-handedness, normal or corrected-to-normal vision, and no individual or familial neurological issues. Hand dominance was evaluated according to the Edinburgh Handedness Inventory [31], and 3 participants were excluded because they were left-handed. The data from the 53 remaining participants (31 women and 22 men; overall mean age 22.09 ± 2.90) were entered in the analysis pipelines. This sample size was in line with previous work on visuomotor interactions, although not computed in advance. For instance, Everard and Gauthier recruited 51 participants to investigate the effect of visual feedback and age on reaching accuracy [47]; Devi et al. used two groups of 45 participants to evaluate the impact of uniocular vision on visuomotor performance [48]; and 40 participants were enrolled by (i) Wood et al. to assess the impact of visual illusions on motor performance [49], (ii) Coudiere et al. [50] and Molier et al. [51] to demonstrate the influence of visual distortion on visuomotor reaching, and (iii) Dexheimer to understand the role of visual input on reaching [52]. All the 53 participants used their right dominant hand to perform the visuomotor task. Prior to the experiment, all participants signed written informed consent forms. The experiment was approved by the local ethics committee and was performed in accordance with the Declaration of Helsinki 2013.

### 2.2. Setup

The setup consisted of participants sitting in front of a desk, using the right dominant hand to hold a computer mouse on the desk. A large screen (95 cm × 55 cm) was positioned in front of them, mounted on a wall, 2.5 m away from the participant. The center of the screen was at a height of 125 cm from the floor. The light in the room was minimized by closing blackout shutters on the windows and by setting artificial lighting to the lowest power. Participants were invited to sit at the desk and were allowed to adjust the height of the seat for comfort. The desk height was 80 cm. A large mouse mat was placed on the desk, denoting the mousing area. The mouse was wireless (60 mm × 99 mm × 39 mm; Logitech M185, Logitech, Zurich, Switzerland), so there were no cables to impede movement. In the computer settings, the mouse sensitivity was set low enough that whole arm movements were required to move the cursor from one side of the screen to the other. The mouse mat was large, with dimensions 90 cm wide by 40 cm long. This size allowed full movement of the mouse without reaching the edges of the mat. Mouse acceleration was deactivated so that the mapping of mouse movement to screen was consistent throughout the experiment.

### 2.3. Experimental Procedure

Building on previous work about visuomotor interactions [50,51,52,53], we set up a visuomotor task based on reaching to draw. The screen background color was set to gray (red = 128, blue = 128, green = 128). The color of the lines drawn on screen was set to black with a thickness set to 1 mm. Before the first trial, a black dot was shown in the center of the screen. This was the starting position. To start the experiment, the participant was asked to click this black dot. Then, the first trial began, with a first white target dot (target 1) appearing elsewhere on the screen. The participant was asked to use the mouse to draw a black line to reach and hit target 1. The starting black dot would disappear as the line started. The hit was registered when the line touched the target. Once hit, target 1 turned black, and the line would disappear. The participant would then click on target 1, and a new white target dot (target 2, trial 2) would appear elsewhere on the screen. At the same time, target 1 would disappear. The participant had to draw a line starting from the position of target 1 to reach and hit target 2 too. Once hit, target 2 would turn black, and target 3 (trial 3) would appear on the screen and should be hit by participants. This would be repeated for each new target till the end of the experiment, which comprised 463 trials (drawing the line from one dot to another one counted as one trial), divided into 160 experimental trials and 303 filler trials. All trials required participants to draw lines between two out of six dots arranged along a circle around the center of the screen but at different distances (Table 1). The mouse position to hit dot #1 was the closest to the participant’s right dominant hand, on her/his right-hand side. To hit dot #2, the mouse had to be placed again close to the participant but on her/his left-hand side. To hit dot #3, the mouse had to be placed on the left-hand side, but farther with respect to dot #2. To hit dot #4, the mouse position was the farthest with respect to the participant’s right dominant hand and on the left-hand side. Also, to hit dot #5, the mouse had to be placed at the farthest possible position with respect to the participant’s hand, but it was on the right-hand side. To hit dot #6, the mouse position was on the right-hand side like for dot #1, but farther away from the participant. These locations were chosen based on the trajectories that could be drawn between them, including short/long, downward/upward, and left/right/straight-directed movements, occurring in all quadrants (proximal/distal, ipsilateral/contralateral with respect to the participant’s right hand (Figure 1). These four trajectories constituted the experimental trials (Figure 2 Left). To avoid learning and to “hide” the experimental trials, the experimental design comprised filler trials too (Figure 2 Right). All participants were able to perform the task, reaching the farthest location (dot) from the participant’s body.

The experimental trial from dot #1 to dot #5 (trajectory A; Figure 1a and Figure 2) implied a long, upward, straight movement. This proximal–distal movement started on the right side of the participant, on the dot closest to the participant’s body, in the bottom-right quadrant. It ended at the furthest node to the body in the top-right quadrant. Trajectory A was entirely on the right side (ipsilateral), and the movement was forward (proximal–distal) along the horizontal plane without any lateral components. The experimental trial from dot #1 to dot #2 (trajectory B; Figure 1b and Figure 2) concerned a short, upward, left-directed movement. This proximal–distal trajectory started on the right side of the participant, on the closest dot to the body in the bottom-right quadrant. It finished slightly further away from the body in the bottom-left quadrant. Trajectory B was primarily across the body to the left (ipsilateral–contralateral) with a lesser forward proximal–distal component. The experimental trial from dot #4 to dot #6 (trajectory C; Figure 1c and Figure 2) induced a long, downward, right-directed movement. This distal–proximal trajectory started on the participant’s left side, on the furthest node from the body, in the top-left quadrant. It ended at the dot closest to the participant’s body in the bottom-right quadrant. Trajectory C was across the body to the right (contralateral–ipsilateral) and back towards the body (distal–proximal). The experimental trial from dot #5 to dot #3 (trajectory D; Figure 1d and Figure 2) required participants to perform a short, downward, left-directed movement. This distal–proximal trajectory started on the participants’ right side, on the dot furthest away from their body, in the top-right quadrant. It finished at the dot that was the second furthest and on the left side of the body in the top-left quadrant (ipsilateral–contralateral). Trajectory D was across the body midline with both the start and finish far from the body.

The four experimental trials further differed in terms of the correspondence between the line drawn and the mouse movement. In 20% of the experimental trials, the line drawn exactly matched the movements of the mouse (undistorted trials). In the remaining 80% of experimental trials, the line would deviate from the mouse movement (distorted trials). In distorted trials, the line drawn would deviate with respect to the movement of the mouse in that, for instance, if the mouse would go straight forward, the line on the screen would go to the left. In these trials the line would be drawn on the screen deviating from the movement by a fixed angle, and the start of the deviation would be from the initial position of the mouse at the start of the trial. There were four possible deviations introduced by the software (−20°, −10°, +10°, +20°), while in undistorted trials the distortion would correspond to 0° (the line accurately maps to the movement of the mouse). For distorted trials, negative values pushed the line toward the left with respect to the direction of the specific trajectory along which the mouse would go, while positive values pushed the line towards the right with respect to the direction of the specific trajectory along which the mouse would go. Any corrections were made by compensating the deviation through exaggerating the mouse displacement while reaching the target dot. In particular, for trajectory A, positive distortions caused the line to deviate to the right, and the associated correction was to be made by moving the mouse towards the left. Negative distortions caused the line to deviate to the left and, therefore, the movements to correct these distortions had to be made towards the right. For trajectory A, the visual cue appeared on the left of the line during positive distortions and on the right of the line during negative distortions. For trajectory B, positive distortions caused the line to deviate upward on the screen (further from the body). Therefore, the correction for positive distortions was made downward, pulling the mouse towards the body. Negative distortions caused the line to deviate downward on the screen (closer to the body). Therefore, the correction for negative distortions was made upward, pushing the mouse away from the body. For trajectory B, the visual cue appeared below the line during positive distortions and above the line during negative distortions. For trajectory C, positive distortions caused the line to deviate downward on the screen (closer to the body). Therefore, the correction for positive distortions was made upward, pushing the mouse away from the body. Negative distortions caused the line to deviate upward on the screen (further from the body). Therefore, the correction for negative distortions was made downward, pulling the mouse towards the body. For trajectory C the cue appeared above the line during positive distortions and below the line during negative distortions. For trajectory D, positive distortions caused the line to deviate upward on the screen (further from the body). Therefore, the correction for positive distortions was made downward, pulling the mouse towards the body. Negative distortions caused the line to deviate downward on the screen (closer to the body). Therefore, the correction for negative distortions was made upward, pushing the mouse away from the body. For trajectory D, the visual cue appeared below the line during positive distortions and above the line during negative distortions.

In line with previous studies about visual distortions of movements [54,55,56,57,58,59,60], this approach resulted in five possible degrees of distortion for each experimental trial, counting the 0° as one possible degree of distortion. Furthermore, while the line was visible in all trials (experimental and filler), the cursor of the mouse was shown in some trials (50%; visually biased trials) but not in others (50%; visually unbiased trials). When visible, the cursor always indicated the true position of the mouse. On this basis, each experimental trial had one degree of distortion (−20°, −10°, 0°, +10°, +20°) and one type of visual input (visually biased, visually unbiased), resulting in 10 possible types of experimental trials. The total number of experimental trials was 160, considering the 4 possible trajectories, 5 possible distortions, 2 possible types of visual inputs, and 4 repetitions for each trajectory.

To minimize the risk of learning due to repetition and to “hide” the experimental trials, the filler trials required participants to perform movements to draw lines between the same dots used for the experimental trials, but in different directions (Figure 2). Like experimental trials, filler trials also included short/long, downward/upward, and left/right, straight-directed movements; the line was visible in all trials, and the cursor was visible in some trials and invisible in others. But, in contrast to experimental trials, the filler trials comprised many more trajectories (Figure 2). Each filler trial was randomly assigned only one out of the five degrees of distortion (−20°, −10°, 0°, +10°, +20°), and the visual input (biased or unbiased) was also chosen at random. Since filler trials were included in the experiment to avoid learning, they were not recorded or included in the analysis.

The presentation order of experimental trials was randomized before the experiment and then presented to the participants. Two consecutive experimental trials were separated by one or two filler trials that were randomly generated. This resulted in the experiment being composed of about one third of experimental trials and about two thirds of filler trials. All participants had the same trials but presented in a different order. The total number of trials was large, so the experimental session was split into two blocks, separated by a rest phase. There was a total of 463 trials for each participant (160 experimental trials and 303 filler trials). The number of 160 experimental trials resulted from the combination of 4 trajectories (Figure 1), 5 degrees of distortion (−20°, −10°, 0°, 10°, 20°), 2 visual inputs (biased, unbiased), and 4 repetitions for each trajectory. Depending on participants’ individual speed in completing the task, each block lasted about 20 min. Participants were instructed that the goal of the trials was to hit the targets with the line (drawing the line from one dot to another one). Prior to the experiment, participants familiarized themselves with the task.

### 2.4. Data Recording

The data from experimental trials was recorded for both the participant’s hand movement (mouse trajectory) and the trajectory that was expressed on the screen as the line. For each data sample, the x and y coordinates were recorded. The screen window in which the lines were drawn had a coordinate system, with the x = 0 and y = 0 with respect to the center of the screen. The screen was calibrated by measuring the length in millimeters of 2 lines (one vertical, one horizontal), each 500 pixels long. This provided a direct mapping of pixels to millimeters on the screen. Data sampling during the experiment was CPU-cycle dependent rather than distance dependent. The length of a trajectory was defined as the distance between two dots as measured on the screen.

### 2.5. Data Preprocessing

The trajectories were normalized and interpolated before data sampling. The normalization of the trajectories was carried out in 3 phases: centering, alignment, and rescaling. Centering: The 2 target dots and trajectory were translated so that the center of the origin node was at the Point of Origin (0, 0). This is the start point. Alignment: The trajectory and destination dot were rotated about the Point of Origin so that the center of the destination dot was at (0, y > 0). The value of y of the rotated destination node depends upon the distance between the 2 nodes. The rescaling comprised interpolation and data sampling. Interpolation: Data recording during the experiment was time dependent and thus was affected by clock cycles, hardware considerations, and, above all, speed of mouse movement. The faster the mouse was moved, the more spread out the sampled data points would be. Participants moved the mouse at different speeds to each other and sped up or slowed down during each trial. This led to an uneven distribution of data points in each trajectory. Therefore, when comparing trajectories, it was important to interpolate the trajectories from the data points to create a data set evenly distributed along the trajectory. Data sampling: The x-value was read from the interpolated trajectory at each millimeter of y from the starting point till the end point.

In addition, the x values for all negative distortions were transformed by multiplying by −1, thus rotating the trajectory 180 degrees about the y-axis. These transformed x values could then be directly compared to the x values for the equivalent positive conditions (−20 vs. 20 degrees, −10 vs. 10 degrees). Finally, according to the data grouping procedure, these data samples were grouped according to the node pair, visual input, and distortion. Then, the grouped samples were put into bins of y-values at 10 mm intervals (0–10 mm, 10–20 mm, etc.). Bin 0–10 was discarded, as were the last bins of the trajectories.

### 2.6. Data Analysis

Following up and building on previous work [53,54], the analysis was focused on the spatial coordinates of the trajectories drawn by the participants. The time taken for each trial was not included in the analysis. To analyze the impact of visual inputs and distortions, we constructed linear mixed models for each bin defined as: x ~ (visual input × distortion) + (1 | subject_id) as independent variables and spatial coordinates as the dependent variable. Bonferroni corrections were applied to post hoc analyses for visual input/distortion interactions. To avoid risks of false positives due to a high number of multiple comparisons, we adjusted the p-value threshold based on the number of bins in each given trajectory (0.05/number of bins). In particular, the adjusted *p*-value thresholds were 0.00147 for trajectory A (34 bins), 0.00192 for trajectory B, 0.00147 for trajectory C (34 bins), and 0.0025 for trajectory D (20 bins). Linear mixed models were calculated using the lme4 version 1.1-35.5 package. Using the emmeans package version 1.10.3, Bonferroni corrections were applied to the main effect and interaction terms resulting from the linear mixed model. All analyses were conducted with the R software (version 4.4.1).

## 3. Results

All the resulting trajectories can be seen in their relative positions in Figure 3.

The results related to trajectory A are represented in Figure 4. For all degrees of distortion, except 0°, the trajectory of the hand movement was significantly different between visually biased and unbiased trials (all *p*_s_ < 0.0001). For distortions of −10°, +10°, and +20°, this significance was valid for the entire length of the movement. The estimate range for the −10° distortion was between 2.138 mm and 7.244 mm. For the +10° distortion, the estimate range was between 1.359 mm and 8.644 mm. For the +20° distortion, the estimate range was between 1.298 mm and 8.845 mm. For the −20° distortion, the significance started at bin 60–70 (*p* < 0.0001) and lasted for the rest of the movement. The range of significant estimates for the −20° distortion was between 2.033 mm and 7.901 mm. The positiveness of the estimates of all distortions means that participants’ movements were more curved in visually biased than unbiased trials. When there was no distortion (0°), although there were significant differences at the start of the trajectory (*p* < 0.0001, estimate = −1.893 mm), the trajectory with the visual cue and the trajectory with no visual cue quickly converged by bin 50–60, and, from then on, there was no difference between the trajectory with the visual cue and that with no visual cue present.

The results related to trajectory B are represented in Figure 5. For the −10°, +10°, and +20° distortions, there were significant differences at the start of the trials, which disappeared by bin 60–70 and reappeared by bin 90–100. For the −10° distortion, the estimate range was between −1.124 mm and −1.290 mm. For the +10° distortion, the estimate range was between −1.090 mm and −1.215 mm. For the +20° distortion, the estimate range was between −1.567 mm and −2.717 mm. Since the estimates for all the significant differences were negative, visually biased trials were associated with more straight movements with respect to visually unbiased trials. After a transition phase starting at bin 60–70, where participants’ performance was independent from visual biasing, starting from bin 150–160, the movements for visual biased and unbiased trials were significantly different (all *p*_s_ < 0.0001). For the −10° distortion, the estimate range was between 1.952 mm and 5.625 mm. For the +10°, the estimate range was between 1.877 mm and 3.984 mm. For the +20° distortion, the estimate range was between 1.976 mm and 3.870 mm. These differences all remained till the end of the trajectory, and all were positive, showing that visually biased trials induced more curved movements with respect to visually unbiased ones. For the −20° distortion, the difference between visually biased and unbiased trials was significant for the entire length of the movement (all *p*_s_ < 0.00019). Since the estimate was positive (the estimate ranged between 1.129 mm and 5.883 mm), the movements for visually biased trials were more curved with respect to those for visually unbiased trials.

The results related to trajectory C are represented in Figure 6. For the −20° distortion, there was no significant difference between movements during visually biased and unbiased trials. For the −10° distortion, with respect to visually unbiased trials, the visually biased trials induced more curved movements for the first half of the trial (up till bin 180–190; *p* < 0.0001; estimate ranged between 2.011 mm and 2.876 mm). Then, the movements for the two types of trials converged and remained the same till the end. For the 0° distortion, visual biasing did not affect participants’ performance. For the +10° distortion, statistically different movements between visually biased and unbiased trials emerged at bin 80–90 till the end of the trial (all *p*_s_ < 0.0001). Since this difference was positive (estimates ranged between 1.661 mm and 4.256 mm), the visually biased trials induced more curved movements with respect to visually unbiased trials. For the +20° distortion, the difference between the movements executed for visually biased and unbiased trails started to be significant by bin 30–40 and remained till the end (all *p*_s_ < 0.0001; estimate range between 1.107 mm and 6.864). The positiveness of these differences showed, again, that the visually biased trials induced more curved movements with respect to visually unbiased trials.

The results related to trajectory D are represented in Figure 7. For the 0° distortion, the difference between the movements associated with visually biased and visually unbiased trails was significant for the entire length of the trial (*p*_s_ < 0.0001; estimate range between 2.026 mm and 3.971 mm). Although the estimated differences were positive, additional care must be taken in interpreting the data in this case. The fact that for the 0° distortion the estimates for both visually biased and unbiased trials were negative means that the calculated estimate values are inverted, and so positive estimate values indicate that the movements for visually biased trials were straighter with respect to visually unbiased trials. Trajectory D is the only one where the presence of the visual cue improves performance and where movements for visually biased and unbiased were significantly different for 0° distortion. For the −10°, +10°, and +20° distortions, at the very start of the trial (from bin 10–20 to bin 50–60), the movements for visually biased trials were significantly straighter than those for visually unbiased trials (*p*_s_ < 0.0002). The estimated values were between −2.885 mm and −0.084 mm for −10° distortion, between −1.201 mm and −1.081 mm for +10° distortion, and between −1.349 mm and −1.082 mm for +20° distortion. Conversely, in the later stage of the trial (starting from bin 90–100), the movements for visually biased trials were more curved with respect to those of visually unbiased trials (*p*_s_ <= 0.0001) till the end of the trial. The estimate values range was between +1.0309 mm and 4.230 mm for −10° distortion, between 1.229 mm and 3.190 mm for +10° distortion, and between 1.196 mm and 4.230 mm for +20° distortion. For the −20° distortion, the movements for visually biased and unbiased trials were not significantly different until bin 150–160, at which point they started to be significantly different (*p*_s_ < 0.0001) with positive estimates ranging from 1.580 mm to 1.070 mm. Like the other three significant distortions, also for −20°, the positiveness of the difference between visually biased and unbiased trials indicated that the movements induced by visually biased trials were more curved with respect to visually unbiased trials.

## 4. Discussion

Overall, the results obtained for all the degrees of distortion except 0° indicated that (i) visually unbiased movements were less influenced by distortions (straighter) than visually biased ones, (ii) the difference between visually biased and unbiased movements was larger when the distortions occurred towards the participant’s body, especially for non-vertical trajectories, and (iii) the only case where movements were straighter in visually biased than unbiased trials is trajectory D, which, interestingly, is also the only case where there is a significant difference between visually biased and unbiased trials at 0° distortion. So apart from the trajectory D, the visual cue consistently worked as a visual distractor, inhibiting correction from distortion [32]. In sum, the differences in movement curvature provide valuable insights into the mechanisms underlying motor control and the role of visual feedback in guiding movements.

### 4.1. Visual Input and Affordance

Compared to visually unbiased movements, visually biased movements were generally more affected by distortions. Visual distortions disrupt the expected visual outcome of a movement and lead the motor system to adjust the movement to accommodate the distortion [44,45]. On this basis, we propose that visually biased trials introduced a stronger disruption with respect to visually unbiased trials. When visual input is unbiased, the motor system can rely more on accurate visual estimates integrated into motor planning [19]. This integration would allow for more precise predictions of movement outcomes, leading to straighter movements. On the other hand, we propose that the stronger disruption brought by visually biased trials would overcome the ability of the motor system to correct visual distortions, resulting in more curved movements. This observation aligns with evidence that motor output is influenced by visual input when closed-loop control is feasible [18]. Fine motor movements like the ones investigated in the present study require complex coordination between the processing of visual inputs and the production of motor outputs [33,34]. Humans rely on visual inputs to achieve optimal motor performance [33,35,36]. Conversely, our findings showed that the presence of a visual cue (visually biased trails) deteriorated the motor performance, in that movements were more curved with respect to unbiased trials. We propose that this apparent contradiction can be explained by considering the weight of affordance in the context of visuomotor interactions. Affordance refers to the possible actions an object offers, based on how it looks and is positioned in the environment. Affordance simplifies the search space of possible actions, thus significantly reducing the computation needed to calculate, for instance, planned routes [37,38]. By this means, affordance improves execution and performance of many tasks. Affordance is very much an interaction between the cognitive processes involved in perceiving what an object represents and the more mathematical mechanics of actual movement [39]. Much work has been carried out on how affordance improves task performance. We propose that the distracting effects of visual biasing are associated with the strong affordance of a mouse cursor. Binding hand movements with visuospatial displacements of a mouse cursor is such an established task in everyday life that when the cursor is in view, it is hard not to follow it, resulting in more curved movements with respect to visually unbiased trials where participants would better focus on the line, not the cursor. In this perspective the mouse cursor was a visual distractor. The appearance of the cursor triggered the affordance response because the cursor, aligned with the user’s actual hand position, directly represents an actionable possibility. This spatial congruence between the hand’s position and the cursor would automatically activate motor responses, as the brain perceives the cursor as an extension of the hand, inviting immediate interaction based on its perceived affordance for action, such as pointing or clicking. In our opinion, the affordance of the cursor negatively affected our task because it automatically triggered a motor response, making participants instinctively want to use it for interaction. This natural tendency conflicts with the task requirement to use the line, not the mouse, to hit the target. The strong affordance of the cursor, representing the actual hand position, can make it harder for participants to shift their focus and control to the line, thereby hindering task performance [40,41,42].

### 4.2. Visual Input and Cognitive Strategy

Reaching tasks like the one used in the present study are frequently used to shed light on the functioning of the visuomotor interactions [43,44,45], showing that altering the quantity of visual input leads to changes in the strategy used to solve the given task. Specifically, individuals tend to depend more on external cues (vision) when visual input is large, whereas they rely more on internal cues (proprioception) when visual feedback is small [46]. In line with this view, we propose that our setup induced participants to focus on external cues (cursor) even if participants were instructed to focus on internal cues (hand movement bound to the line). We suggest that such a strong influence of the visual cue has links to the so-called Simon effect. The Simon effect suggests that motor responses are faster and more accurate when the spatial location of a stimulus corresponds with the required response, even if that location is irrelevant to the task. In our setup, the cursor was congruent with the hand’s real position and, therefore, could activate a spatial response bias. Participants might naturally gravitate toward using the cursor to hit the target because it is spatially congruent with their hand, reinforcing the affordance effect. This spatial alignment would make participants more likely to focus on the cursor rather than the line, making it harder for them to focus on the line, which is the actual task. Thus, the Simon effect may amplify the challenge by reinforcing participants’ inclination to interact with the cursor, which distracts them from accurately using the distorted line to hit the target. When the cursor was present, visual cues would guide movements, similar to how irrelevant spatial information influences responses in the Simon effect, causing automatic motor activations based on visual input. When the cursor was absent, the reliance shifted from external visual guidance to internal cues like proprioception (the sense of body position and movement). In this situation, motor responses are guided more by the body’s internal awareness, and the automatic spatial-motor activation observed in the Simon effect may be diminished because spatial information from vision is no longer dominant. These exchanges between externally and internally cued strategies, in combination with the affordance response, would enhance the tendency to respond to the spatially congruent cursor, leading to reduced accuracy in using the distorted line [40,41,42].

### 4.3. Visual Input and the Body

The difference between visually biased and unbiased movements was more pronounced when distortions occurred towards the participant’s body, especially for non-vertical trajectories. This finding can be related to the crucial role of visual and proprioceptive information in visuomotor adaptation [40]. We propose that when distortions are directed towards the body, they likely create a greater conflict between visual and proprioceptive inputs. This conflict would arise because the visual system and the proprioceptive system (which provides information about the position and movement of the hand) must work together to guide accurate movements. When these systems receive conflicting information, the motor system faces increased difficulty in integrating these inputs to produce a coherent movement plan [61]. According to this view, the reliance on proprioceptive feedback would be particularly critical for non-vertical movements, due to more complex motor planning/execution and more precise coordination between visual and proprioceptive inputs. Thus, distortions towards the body would exacerbate the challenge of maintaining accurate movement paths, as the motor system must constantly adjust to reconcile the conflicting information. This increased difficulty would be in line with the larger difference observed between visually biased and unbiased movements under these conditions [40]. Since visual distortions disrupt the expected visual outcome of a movement, leading to adjustments in the movement trajectory [44,45], we propose that the larger influence of visual bias on distortions towards than away from the body may be due to the fact that visuomotor coupling is strengthened in the space immediately surrounding the body, known as peripersonal space. This interpretation is supported by previous evidence that objects near the body are more likely to engender sensations [62], visual signals from near-body space converge in motor brain regions [63], and the spatial stimulus-response mapping near the body is increased, as demonstrated by a stronger Simon effect when hands are proximal to stimuli [64]. In our view, the more robust visuomotor coupling in near-body space can explain why the difference between visually biased and unbiased movements is larger when the distortions occurred towards than away from the body. Since trajectory D was the farthest with respect to the participant’s body, its exception further supports the importance of the body as the reference frame for visuomotor interactions. It indicates that the motor system may develop near- and far-space-dependent strategies to anticipate and counteract distortions, leading to more accurate movements under certain conditions. Interestingly, manipulations of the visual input in the space near the body have proven sufficient to induce changes at the cognitive level too [65,66], and sensory breakdown affects visuomotor integration in peripersonal space [67]. Overall, our findings emphasize the complexity of visuomotor integration and the adaptability of the motor system in response to varying sensory inputs. The present study highlights the importance of the interplay between visuomotor interactions and the reference frame, with a particular focus on the body. The larger impact of distortions towards the body underscores the complexity of visuomotor integration, demonstrating how the motor system body-dependently adapts to conflicting visual information.

### 4.4. Applications

The findings that visually unbiased movements are generally straighter and less influenced by distortions compared to visually biased movements may have several practical applications. In rehabilitation, they might be beneficial for the development of therapies for visuomotor impairments like optic ataxia, a disorder characterized by impaired reaching to visual targets in the absence of basic motor impairments [68,69,70]. For instance, based on evidence that a temporal delay in pointing tasks improves the performance of patients with optic ataxia [71], it could be proposed to design exercises customized on the effects of visual distractors that take into consideration the role of the presence/absence of visual input, the affordance associated with specific types of visual inputs, and the body-related reference frames for visuomotor interactions. These three aspects might be exploited for technological developments, such as robotic prosthetics and human–computer interaction. Typically, prosthesis users tend to perform reaching based largely on visual attention [72], but sensory feedback and training can reduce the reliance on only vision [73]. Showing the role of visual feedback on visuomotor performance, the present study could help the design of prosthetics and assistive devices to better embed the impact of visual biases on motor performance, thereby achieving more natural and precise movements. In a similar vein, building on evidence that stroboscopic visual training (intermittent absence of visual input) improves visuomotor performance [74,75], sports training coaches may use the knowledge brought by the present study (presence/absence of visual input, affordance, body reference frame) to develop training programs that help athletes optimize the integration of visual input and motor output, leading to improved performance.

### 4.5. Limitations and Perspectives

It might be argued that our task could be affected by the anthropometric characteristics of the participants. While we note that all participants were able to perform the task and that the absence of anthropometric measures is in line with most of previous studies [53,76,77,78,79,80], we acknowledge that the possible role of anthropometric characteristics deserves particular attention and can be the focus of future investigations.

In addition, while the range of distortion was based on previous studies, it may be argued that the present study assessed the impact of a limited range of visual distortions. We acknowledge that testing a wider range of distortions could provide important information about how the relationship between movement execution and visual input varied as a function of the degrees of their coherence [81]. An in-depth analysis of this topic could be the focus of future experiments.

Finally, the present experiment comprised trajectories of different lengths. While this was executed on purpose in order to minimize the risk of learning, we acknowledge that further studies may take into account the impact of trajectory length of visuomotor interactions.

## 5. Conclusions

Many everyday tasks rely on visual inputs to enhance motor accuracy and reduce errors. Reaching tasks provide an ecological approach to investigate these movements and analyze visuomotor interactions. This study examined how the presence or absence of visual input related to hand position affects motor performance during a reaching task. The results revealed that when the hand-related visual cue was visible and separate from the task-related visual cue, movements made in response to spatial distortions were less accurate compared to when the visual cue was not present. We interpret these results in the context of motor affordance regarding visual cues and the shifts in reliance on internal versus external cues during movement execution.

## Figures and Tables

**Figure 1 brainsci-14-01270-f001:**
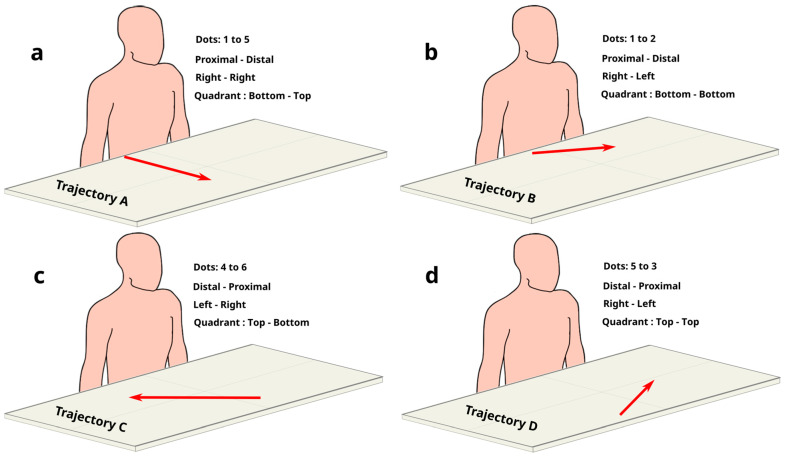
Experimental trials are represented by red arrows. (**a**) Participants performed movements away from the body, ipsilateral to the dominant hand, and towards the space farther from the body (Trajectory A), (**b**) away from the body, contralateral to the dominant hand, and towards the space closer to the body (Trajectory B), (**c**) towards the body, contralateral to the dominant hand, and towards the space closer to the body (Trajectory C), and (**d**) towards the body, contralateral to the dominant hand, and towards the space farther from the body (Trajectory D). See also Table 1 and Figure 2.

**Figure 2 brainsci-14-01270-f002:**
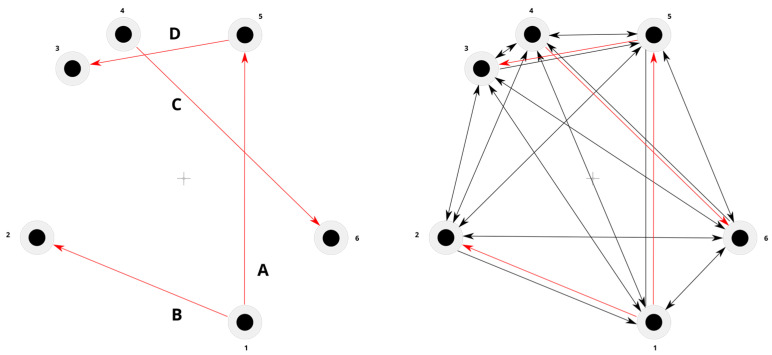
(**Left**) Experimental trials (red arrows) were selected to let participants perform short/long, downward/upward, left/right/straight-directed movements across all quadrants. (**Right**) Filler trials (black arrows) were added to “hide” the experimental trials. Numbers represent the selected dots. Letters represent the trajectories described in Figure 1 and Table 1.

**Figure 3 brainsci-14-01270-f003:**
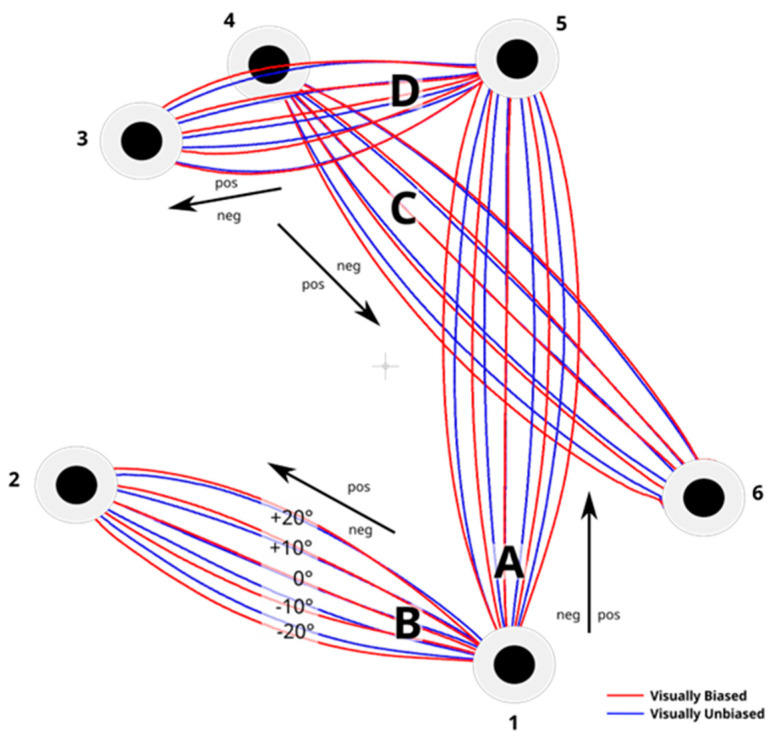
Data of experimental trials showed where the trajectories of visually biased and visually unbiased trials were different as a function of the degrees of distortion. Each line represents the average x value across all participants for a specific distortion (−20°, −10°, 0°, +10°, +20°). Red lines show the trials where there was visual biasing with the cue present. Blue lines show trials where there was no visual biasing (no cue). Thus, for each trajectory there are 10 lines shown (5 red, visually biased lines, and 5 blue, unbiased lines). For illustration purposes, the degrees of distortions are shown only on trajectory B. Numbers represent the selected dots. Letters represent the trajectories as described in Figure 1 and Table 1.

**Figure 4 brainsci-14-01270-f004:**
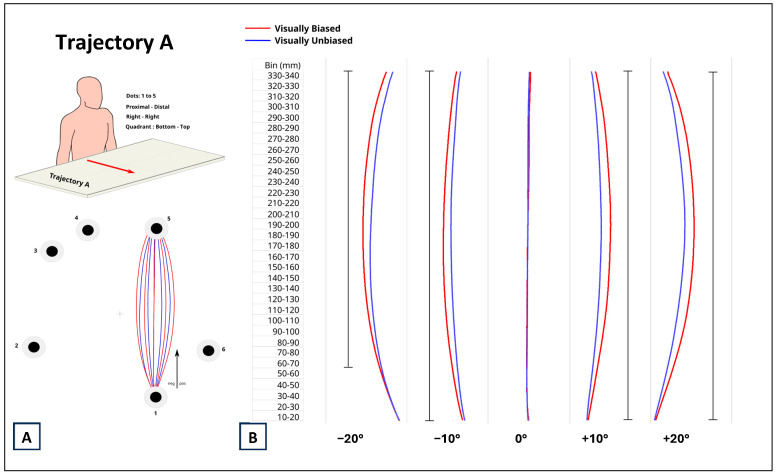
(**A**) Data from trajectory A corresponded to proximal–distal, ipsilateral, bottom–top reaching movements. Numbers represent the selected dots as described in Figure 1. (**B**) Comparison between reaching movements performed in presence (visually biased; red curves) and absence (visually unbiased; blue curves) of a visual distractor (computer mouse), separately for each degree of visual distortion (−20°, −10°, 0°, +10°, +20°). The full black vertical line next to each pair of curves indicates where along the length of the reaching task (Bin mm) the difference between visually biased and visually unbiased movements was significant.

**Figure 5 brainsci-14-01270-f005:**
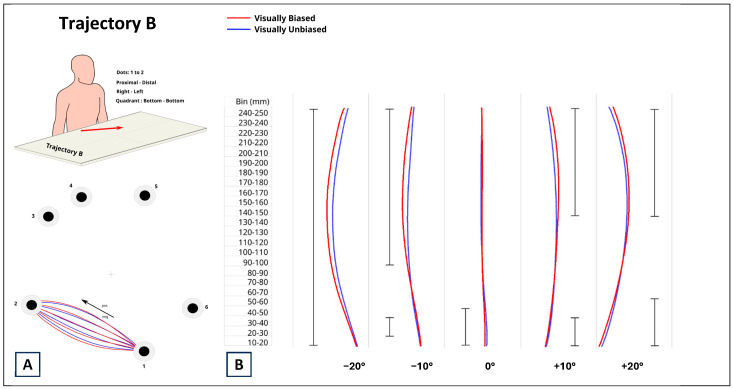
(**A**) Data from trajectory B corresponded to proximal–distal, contralateral, bottom–bottom reaching movements. Numbers represent the selected dots as described in Figure 1. (**B**) Comparison between reaching movements performed in presence (visually biased; red curves) and absence (visually unbiased; blue curves) of a visual distractor (computer mouse), separately for each degree of visual distortion (−20°, −10°, 0°, +10°, +20°). The full black vertical line next to each pair of curves indicates where along the length of the reaching task (Bin mm) the difference between visually biased and visually unbiased movements was significant.

**Figure 6 brainsci-14-01270-f006:**
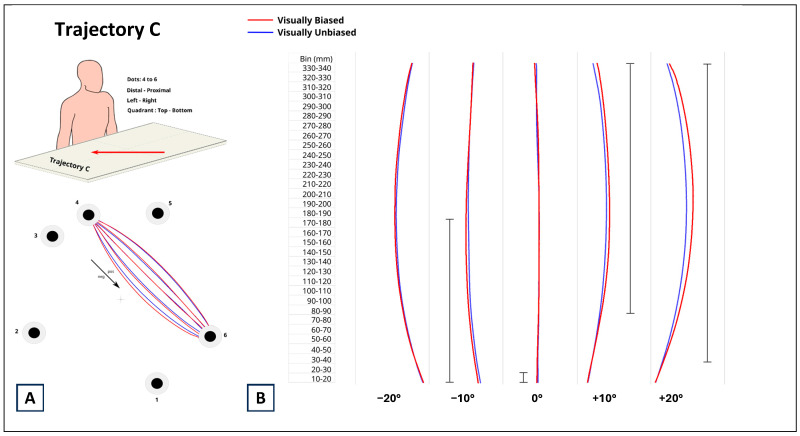
(**A**) Data from trajectory C corresponded to distal–proximal, contralateral, top–bottom reaching movements. (**B**) Comparison between reaching movements performed in presence (visually biased; red curves) and absence (visually unbiased; blue curves) of a visual distractor (computer mouse), separately for each degree of visual distortion (−20°, −10°, 0°, +10°, +20°). The full black vertical line next to each pair of curves indicates where along the length of the reaching task (Bin mm) the difference between visually biased and visually unbiased movements was significant.

**Figure 7 brainsci-14-01270-f007:**
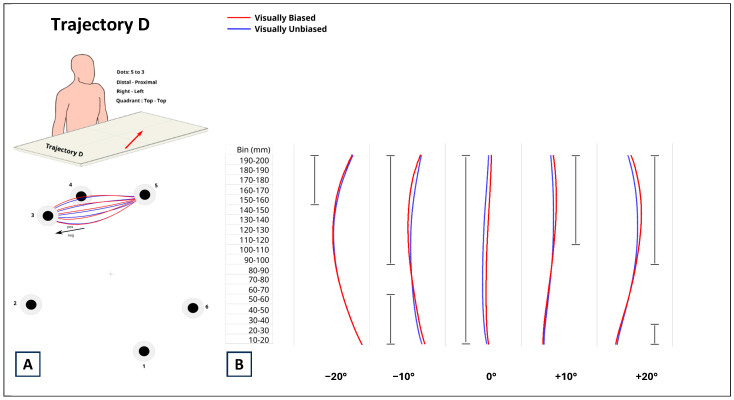
(**A**) Data from trajectory D corresponded to distal–proximal, contralateral, top–top reaching movements. (**B**) Comparison between reaching movements performed in presence (visually biased; red curves) and absence (visually unbiased; blue curves) of a visual distractor (computer mouse), separately for each degree of visual distortion (−20°, −10°, 0°, +10°, +20°). The full black vertical line next to each pair of curves indicates where along the length of the reaching task (Bin mm) the difference between visually biased and visually unbiased movements was significant.

**Table 1 brainsci-14-01270-t001:** Experimental trials concerned four directions of movements (Trajectory), each connecting two points (Dots), positioned at short and long distances as measured from the screen (Length), starting (Start Quad) and ending (End Quad) in the four possible quadrants and towards or away from the participant’s body (Direction), on the same (Ipsi) or different side (Contra) with respect to the participant’s right dominant hand (Side).

Trajectory	Dots	Length (mm)	Start Quad	End Quad	Direction	Side
A	1 to 5	351	Bottom Right	Top right	Proximal–Distal	Ipsi–Ipsi
B	1 to 2	269	Bottom Right	Bottom Left	Proximal–Distal	Ipsi–Contra
C	4 to 6	351	Top Left	Bottom Right	Proximal–Distal	Contra–Ipsi
D	5 to 3	210	Top Right	Top Left	Distal–Proximal	Ipsi–Contra

## Data Availability

The data presented in the present study are openly available in Zenodo at https://zenodo.org/records/14007005 (accessed on 29 October 2024).

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
