# Peer review of "Body-Related Visual Biasing Affects Accuracy of Reaching"

_brainsci, 2024, doi:10.3390/brainsci14121270_

Round 1
Reviewer 1 Report
Comments and Suggestions for Authors
Review of "Vision Dependent Kinematics"
The authors use a large-scale line drawing task to measure the effect of visual feedback under conditions of altered mouse mapping.
The task is somewhat unusual in that the surface for drawing is nearly as large as the monitor on which the target it to be drawn, involving large arm movements. Data from 53 participants is included.
1. I have one methodological question concerning the mouse:
For most mouse interfaces, cursor motion is not a simple remapping of mouse motion because higher mouse velocity over the same physical distance produces a larger movement of the cursor. It was not clear to me whether or not the this was true of the mouse settings on the computer in this experiment. This detail does not invalidate the experiment, but it would be helpful if the authors are clear about whether their mouse accelerated in this typical way rather than acting as a tablet.
2. Theoretical concern. What is the experiment about? The experimental manipulation is stated to involve the presence of visual information about the cursor position (visually tracked vs not visually tracked). But, on most trials, the relevant cursor position is also given by the visible line that the participant is drawing isn't it? Thus the manipulation that the experimenters describe seems to me to involve introducing a conflicting, irrelevant position cue (the cursor) that on most trials (all the filler trials plus 1/5 fo the experimental trials) is redundant with the line. Thus, the difference observed between trials with the cursor position and those without is not really just the presence of visual tracking (which is always present), but a specific kind of misleading feedback (a true cursor position that deviates from the drawn line) that the participants simply seem to get more distracted by.
2a. Minor: Signal vs. noise? I think calling the intermixed trials "filler" and the manipulated trials "experimental" trials would be more consistent with common practice.
3. So what is the theoretical question being addressed? Is it a question of whether unhelpful feedback is more misleading than simple motor remapping? Is it the fact that when visual feedback (the line) and proprioceptive information disagree, better corrections are made than in the case when an added visual cue supports the (unhelpful) proprioceptive feedback?
4. In short I am not sure I understand what the question is here given that all conditions involve continuous visual tracking.
5. Analysis: The tables of significance valued for each cm of path seem somewhat unhelpful. I'm not sure what to suggest as a better method of presentation, but what is the theoretical value of a table of p-values using 1 cm interpolated points (clearly not independent of one another)? Couldn't one just specify the portion of the path (e.g., "from 10% until 95% of the way along the path, the offset cursor representation caused reliably greater deviation from a straight path than using the line alone." The images show the deviations already, and the table of p-values just seem like they provide the wrong sort of false precision of heavily massaged data.
6. Overall, the empirical results here seem interesting, but neither the theory (visual tracking is always present!) nor the analysis (treating trajectories as series of independent points) seems helpful. I've made suggestions about improving both:
a. present a more succinct analysis rather than huge tables of p-values, b. consider whether the less correction occurs because the cursor information simply conflicts with the line being drawn.
Reviewer 2 Report
Comments and Suggestions for Authors
The authors present an interesting topic by investigating how the presence or absence of visual input can influence hand motor performance. The methodology needs some improvements because the description of the procedures seems a little confusing, and some aspects need to be clarified. Furthermore, I am concerned about the statistical analysis and how the results were presented. The discussion requires further exploration of the results.
Introduction
- In line 35, the authors start their references in number 3, but there is no indication of references in the previous sentences, so they must rectify them.
- The authors should further explore the visual system's role in sensorimotor integration and motor response.
- The authors should add studies that have analyzed the association between visual input and motor performance and make a connection between what already exists in the literature and the novelty of the present study.
- Please clarify the aim of the study, adequately justify its conduct, and present the research hypotheses.
Material and Methods
Point 2.1
- Present the eligibility criteria and anthropometric characteristics of the sample in detail.
- How was the sample recruited? Did an ethics committee approve the study? Was any sample calculation performed? Present this data, please.
Point 2.2.
- State the height of the desk and the type of mouse used, as there are different formats and sizes of mouse.
- If you have images of the setup or a mouse, add them for better visualization of the study setup.
- Did everyone do the tests with their right hand?
Point 2.3
- Authors must mention whether they relied on previous studies to plan their experimental procedures or whether it is an exploratory procedure.
- Lines 95 to 97: For better understanding by the reader, clarify. Did each participant only perform the trial once? What do signal trials represent? What do you consider noise trials? And how did you arrive at number 303?
- Line 99: “With respect to the vertical”. Vertical in relation to what? What is the point of origin? Are the angles formed based on the mouse's position, the various points, and the center of the computer screen? Explain the reference for determining angles.
- Lines 107, 122, 129, 142, and 155, 193, 251, 257, 276, 299, 317: “(Error! Reference source not found.)”. Clarify or rectify if it is a data entry error.
- In Figure 2: To make it easier for the reader to interpret the figure, label which trajectory each letter corresponds to.
- Indicate the tables and figures 4 and 5 in the text. It should not just present the tables or figures on the page, but also reference them in the text.
- It is not very clear how they assess motor performance. For example, is it through the assessment of motor precision or reaction time?
- Line 172 to 174: “In some trials (20%) the line drawn exactly matched the movements of the mouse (undistorted trials). In other trials (80%) the line would deviate from the mouse movement (distorted trials)” and line 184: “trials (50%; visually tracked trials) but not in others (50%; visually untracked trials)”. When the authors mention this, it is already about results, so this type of information should be in the results and not in the methodology.
- Lines 177 to 178: “These distortions could introduce one of four possible biases (-20°, -10°, +10°, +20°), while in undistorted trials the distortion would correspond to 0°.” How did you arrive at the bias angles? Was it the average error of each distortion determined at the end of the trial?
- Lines 199 to 200: “experimental session was split into 2 blocks.” Please explain what each block consists.
- Line 206: “For each data sample, the x and y coordinates were recorded.” Indicate what the x and y axis relate to.
Point 2.5
- Was data processing based on previous studies? If not, how can you assess whether the data processing is the most appropriate?
Point 2.6.
- Was the normality of the data verified? Did they use parametric or nonparametric tests? What is the level of statistical significance?
Results
- In Figure 5: Label what the blue and red lines represent.
For example, in trajectory B, there are 5 lines. Does this mean that for each trajectory, each participant performed 5 repetitions?
- Rectify the number of tables.
Points 3.2., 3.3. and 3.4.
- If the significance level considered was p<0.05, there are values ​​in the table with statistical significance that are not highlighted.
Discussion
- Do you have data on which eye is dominant? If so, it would be interesting to discuss the results.
- The results of the present study should be further explored and analyzed according to the available scientific evidence.
- The authors can add information on how to apply the results of this study in the discussion section (e.g., theoretical contribution and practical contribution).
- Insert the limitations and suggestions for future lines of research in the discussion.
References
Rectify the references because the manuscript does not mention all the authors referenced in this section. For example, 46 references are mentioned in this section, and in the manuscript, it only appears up to reference 42.
Reviewer 3 Report
Comments and Suggestions for Authors
The manuscript reports of a study to investigate how the availability of visual input impacts the motor action in a mouse-tracking task. The results showed that the tracking was more accurate in the condition without visual input.
The study appears to be carried out and analyzed in an appropriate way, but the manuscript would need to be revised to clarify a number of issues. The introduction needs more motivation for the present study. Results of related studies could be introduced more (e.g., references 43-45). What knowledge gap does the current study address, more specifically? Ending the introduction with a conceptual outline of the study, and perhaps also hypotheses, would aid the understanding. The experimental design should be clarified in the methods section (independent and dependent variables). It could be clarified whether the order of trajectories was controlled and whether participants were given a warm-up phase.
Minor issues:
I would like a more informative title.
The results section is rather difficult to read - is it possible to present it in a simpler and clearer way?
Line 64: Sentence starting with a number, incorrect capitalization of "Healthy"
and other minor typing and formatting errors. Also Table and Figure numbers are incorrect.
Table 1 (Line 255): In the heading, "P <" should be "P ="?
Figure 5-9: an explanation of the red and blue colours is missing.
Round 2
Reviewer 1 Report
Comments and Suggestions for Authors
I think the authors have done a good job addressing the previous reviews. The contribution is clearer as a result. The theoretical motivation of the study seems unclear to me. This is an empirical study comparing normal visual drawing feedback in a perceptuomotor task with an angular offset with having additional irrelevant feedback (the cursor) that reinforces current proprioceptive expectations. The generally weaker performance with that additional feedback could be due to many factors including divided attention, visual distraction, increased conflict cues, etc., but the study seems to be well done, and this draft seems sufficiently clear to me concerning the paradigm. I have no further suggestions.
Reviewer 2 Report
Comments and Suggestions for Authors
Authors improve their manuscript. However, I offer some comments:
Just being based on the sample number used in other studies does not represent a sample size calculation.
The authors present the mean age and standard deviation for females but not for males. This information should be available to both genders.
In lines 562 to 577, the authors could add some references that corroborate the applications.
Reviewer 3 Report
Comments and Suggestions for Authors
I thank the authors for the revision and conclude that all issues I raised in the initial review have been adequately addressed.
Author Response
We thank the reviewer for the constructive feedback that helped us improve the manuscript.